# GWAS for Early-Establishment QTLs and Their Linkage to Major Phenology-Affecting Genes (*Vrn*, *Ppd,* and *Eps*) in Bread Wheat

**DOI:** 10.3390/genes14071507

**Published:** 2023-07-24

**Authors:** Md. Farhad, Shashi B. Tripathi, Ravi P. Singh, Arun K. Joshi, Pradeep K. Bhati, Manish K. Vishwakarma, Uttam Kumar

**Affiliations:** 1Bangladesh Wheat and Maize Research Institute (BWMRI), Dinajpur 5200, Bangladesh; md.farhad@bwmri.gov.bd; 2TERI School of Advanced Studies, Vasant Kunj, New Delhi 110070, India; shashi.tripathi@terisas.ac.in; 3International Maize and Wheat Improvement Centre (CIMMYT), Carretera México-Veracruz Km. 45, El Batán, Texcoco C.P. 56237, Mexico; r.singh@cgiar.org; 4Borlaug Institute for South Asia (BISA), New Delhi 110012, India; a.k.joshi@cgiar.org (A.K.J.); pk.bhati@cgiar.org (P.K.B.); m.vishwakarma@cgiar.org (M.K.V.)

**Keywords:** agro-morphological traits, early planting, GWAS, ideotype design, phenology

## Abstract

Farmers in northern and central Indian regions prefer to plant wheat early in the season to take advantage of the remaining soil moisture. By planting crops before the start of the season, it is possible to extend the time frame for spring wheat. The early-wheat-establishment experiment began in the 2017 growing season at the Borlaug Institute for South Asia (BISA) in Ludhiana, India, and, after three years of intensive study, numerous agronomic, physiological, and yield data points were gathered. This study aimed to identify wheat lines suitable for early establishment through an analysis of the agro-morphological traits and the genetic mapping of associated genes or quantitative trait loci (QTLs). Advancing the planting schedule by two–three weeks proved to be advantageous in terms of providing a longer duration for crop growth and reducing the need for irrigation. This is attributed to the presence of residual soil moisture resulting from the monsoon season. Early sowing facilitated the selection of genotypes able to withstand early elevated temperatures and a prolonged phenological period. The ideotype, which includes increased photo-growing degree days for booting and heading, as well as a longer grain-filling period, is better suited to early planting than timely planting. Senescence was delayed in combination with a slower rate of canopy temperature rise, which was an excellent trait for early-adapted ideotypes. Thus, a novel approach to wheat breeding would include a screening of genotypes for early planting and an ideotype design with consistent and appropriate features. A genome-wide association study (GWAS) revealed multiple QTLs linked to early adaptation in terms of the yield and its contributing traits. Among them, 44 novel QTLs were also found along with known loci. Furthermore, the study discovered that the phenology regulatory genes, such as *Vrn* and *Ppd*, are in the same genomic region, thereby contributing to early adaptation.

## 1. Introduction

The prevailing notion is that extending the crop duration for cultivation by a designated period may offset the decline in crop productivity resulting from the rise in seasonal temperatures. This has been exemplified by a few varieties of wheat in India, which manifested accelerations of 4–9 days in the onset of flowering [1]. By sowing spring wheat before the standard planting period, it may be feasible to extend the duration of the planting intervals. For instance, initiating the planting process from one to two weeks ahead led to a significantly extended crop growth period, accompanied by the elimination of one irrigation requirement owing to the presence of residual moisture after the monsoon season. However, it was expected that an unfavorable year would lead to vulnerability and eventually reduce crop productivity [2]. Warmer temperatures in the early stages of the growing season accelerate the occurrence of heading and anthesis during grain filling, while cooler temperatures later on lead to a delay in maturity. Irrespective of the constancy of the mean seasonal temperature, there is a positive correlation between a decrease in temperature and a reduction in yield, while an increase in yield is observed with a temperature rise [3]. Because the monsoon season is the wettest, farmers in many northwestern and central Indian regions would prefer to sow wheat earlier in the season to capitalize on the available soil moisture. If wheat genotypes are sown earlier than the recommended planting date, then they might experience elevated temperatures during root development, leaf development, and emergence, significantly in advance. From seeding to emergence, the optimal temperature range is between 20.4 and 23.6 °C, with the highest temperatures recorded reaching between 31.8 and 33.6 °C [4]. Root development during the vegetative stage thrives at an ideal temperature of 20 °C, whereas shoot growth prefers significantly lower temperatures [5]. Based on a research study, it was determined that the most favorable temperature range for the emergence of leaves is between 21.3 and 24.3 °C [6], whereas it is 22 °C according to another study [7].

There is a lack of understanding about how genes regulate the early stages of heat tolerance and the adaptive response. Many of the key genes involved in adaptation have been identified in recent years. These include genes involved in the vernalization reaction (*Vrn*), photoperiod sensitivity (*Ppd)*, and “earliness per se” (*Eps*) in terms of the growth rate [8,9,10,11]. However, there are significant differences in our understanding of the implications of potential allele combinations for environment-specific adaptations. On the one hand, the *Vrn* genes regulate the distinction between spring and winter wheat by specifying the number of chilling hours necessary for the wheat plant to initiate flowering. On the other hand, the *Ppd* genes are crucial in postponing the flowering time during spring, once the vernalization requirement has been met. The effects of the *Eps* loci may enable more nuanced adjustments to the plant’s life cycle, facilitating regional adaptation. Several areas possibly harboring *Eps* genes have been mapped out by Quantitative trait locus quantitative trait loci (QTL) investigations [12,13]. Allelic heterogeneity appears to be prevalent in various germplasms for specific QTLs, including those found on chromosomes 4B, 6A, and 7D. The genes *Vrn*, *Ppd*, and *Eps* also contribute to epistatic interactions [14,15]. As a result, numerous combinations of alleles are likely to influence the regulation of growth patterns and optimal adaptation to particular climates.

Planting wheat before the season starts has several adverse effects on the plant’s growth and development. Early planting may increase the risk of unproductive vegetative growth, reduce tillering, and make plants more susceptible to drought in dry conditions. The high soil temperatures in the planting stage may reduce the germination percentage and coleoptile length. Furthermore, the deeply planted seed may not have a long enough coleoptile to split through the soil surface, leading to lower emergence and less adequate stand establishment. The early planting of traditional and short-duration varieties, with photoperiod insensitivity, leads to early heading and maturity, resulting in yield reduction. Therefore, selecting advanced breeding lines suitable for early planting may be difficult but it is not impossible, as diverse germplasms can give these variations to show a better adaptive capacity in early planting.

The mapping of agronomic and phenological traits is a continuous process in wheat development. QTL mapping has been widely used in wheat breeding to understand the probable genetic control of loci, genes, or even genome segments in biotic and abiotic stress adaptation or resistance. Early wheat establishment was initiated at the BISA in Ludhiana, India, in the 2017 season, and, after three years of extensive research, several data points were generated. The early-planted wheat lines had three theoretical types of adaptation: (a) suitable for early- as well as timely-planted conditions, (b) suitable for early planting but not suitable for timely planting, and (c) suitable for timely planting but not suitable for early planting. This adaptation expresses phenotypically with variation in morpho-physiological traits, and genotypically with the effect of different genes. Diseases may also be a limiting factor for early-established plants.

With this perspective in mind, the current study focused on finding wheat lines for early establishment by analyzing agro-morphological features and mapping genes or QTLs associated with early-adaptation features. A few agro-morphological traits supporting genotypes for excellent performance in early planting have already been identified in our previous study, along with the ideotype selection procedure [16]. The major research questions to identify QTLs associated with adaptation to early planting for this study are as follows:(a)Can we identify the major QTLs important for the early establishment of wheat?(b)What are the QTLs associated with higher performances in both planting conditions?(c)Are those QTLs linked with major genes underlying adaptation to early planting?

## 2. Materials and Methods

### 2.1. Phenotyping

Three distinct sets of advanced spring wheat (*Triticum aestivum* L.) breeding lines were evaluated during multiple wheat seasons from 2017 to 2020. Early sowing was performed at the Borlaug Institute for South Asia (BISA) in Ludhiana, Punjab, India, as part of the study’s hypothesis to extend the wheat-growing window. As Punjab is India’s high-yield-potential wheat-growing zone, early planting should increase wheat production and have a significant impact on the region’s agricultural system. The germplasm lines used in this study were developed at CIMMYT, Mexico, and designated as a South Asia Bread Wheat Genomic Prediction Yield Trial (SABWGPYT). Six hundred genotypes, including checks, were planted in an α lattice design [17] with two replications in the first week of November, referred to as a timely (or normal) seeded experiment. In contrast, the same genotypes were planted for early planting, around three weeks earlier than the regular timetable, with a range of 17 days in season 1, 24 days in season 2, and 23 days in season 3. Each replicated block was split into six sub-blocks, each with ten plots. Each plot had six rows and measured 1.32 m × 3.80 m. There was no difference in the seed rate (50 g per plot) between planting early and planting on time. Standard agronomic procedures suggested for the region were used to manage field experiments. Five irrigations were administered starting 21 days after sowing, with additional irrigations provided throughout a 3–4-week period, depending on weather circumstances. The following proportions of fertilizer were applied per hectare of land: 150N:60P:40K kg.

Several morpho-physiological traits were assessed in the field for field phenotyping. These traits were categorized according to their type. Phenological features include all of the natural events that occur repeatedly throughout the wheat life cycle. For instance, plant stature traits quantify the heights of the plants’ different portions. Physiological attributes include a variety of field-observable physiological activities in wheat that can be quantified using high-throughput phenotyping. Additionally, we gathered data on yield and yield-contributing traits for this investigation. To minimize time and effort, the FieldBook App (https://www.phenoapps.org/apps/; accessed on 27 April 2023) developed by Poland Lab at Kansas State University in the United States was utilized for high-throughput phenotyping [18].

#### 2.1.1. Plant Phenology

The booting stage is characterized by a notable enlargement of the developing head within the sheath of the flag leaf. Once the flag-leaf sheath begins to unwrap and the first awns emerge, the procession is finished. Days to booting were calculated using the date at which 50% of the plants in a plot completely emerged during the booting stage (DTB). The heading date is defined as the point at which the tip of the head emerges from the flag-leaf sheath before the head completely emerges but before it begins to bloom. When 50% of the plants in a plot had fully grown into the head stage, the days to head (DTHD) were determined. The days between the booting and heading stages were referred to as booting to heading days (BTH). Days to maturity (DAYSMT) were determined as the number of days between sowing and physiological maturity. The point of ultimate maturity was determined as the moment when the peduncle exhibited a 50% loss in its green pigmentation. The duration between the heading stage and physiological maturity, when the grain-filling time is taken into account, is commonly known as the heading to maturity days, or the grain-filling duration (GFD).

#### 2.1.2. Plant Stature

Plant height (PH) was taken by grasping a clump of spikes and measuring the distance from the ground to the tips of the most representative spikes (excluding the awns). The measurement of the plant’s height up to the spike base is denoted as the height up to the spike (HUS). This height suggests the vegetative height of the plant. The spike length (SpkLng) is the height of the spike from its base to its tip. The leaf-blade length was measured as the flag-leaf length (FLGLFL). The flag-leaf width (FLGLFW) was estimated by folding the leaf in half along its length and measuring its width at the crease. The following formula, derived from prior studies, has been proposed for calculating the flag-leaf area (FLGLFA): [19,20]: FLGLFA=FLGLF×FLGLFW×0.75. The mean plant height and flag-leaf traits of five randomly selected plants were measured during the grain-filling stage and expressed in centimeters.

#### 2.1.3. Physiological Traits

Visual assessment was conducted to determine early ground cover (EGC) or rapid canopy closure by observing the percentage of soil covered by green tissue in each plot. This assessment was performed by viewing the plots at a 45-degree angle to the vertical axis. Typically, EGC is practiced during Zadok’s Growth Stage 15, which corresponds to the 5-leaf stage of plant development. At this stage, the visibility of the ground beneath the plant canopy is minimal in some plots. The day on which at least one plot appeared to be completely covered by the plant canopy was chosen as the reference day for the EGC assessment. Comparing this 100% canopy cover, we gave a visual score to each plot as a percentage for EGC, as per the guideline proposed by previous studies [21]. An infrared thermometer (IRT) was utilized to determine the temperature of the canopy. As the temperature of canopies increases, they emit long-wave infrared radiation. The IRT detects the emission and subsequently converts it into an electrical signal, which is then represented as temperature. It is imperative to utilize the thermometer appropriately to obtain precise outcomes. We adhered to the procedure outlined by Julian Pietragalla (2012) [22]. The canopy temperature increases with the time from heading to maturity as the air temperature increases. The genotype consistently maintains its cooler canopy over the grain-filling period, which is considered a stable genotype for heat tolerance. The slope of the canopy temperatures over time were taken as the canopy temperature increasing rate (CTIR).

The normalized difference vegetation index (NDVI) is also extensively used to measure vegetative greenness and the canopy photosynthetic size at ground level, as well as from low, higher, and satellite altitudes. The field-portable NDVI sensor is used to measure a crop’s NDVI values at a high resolution to classify the canopy for the leaf-area index (LAI) and green-area index (GAI), biomass, and nutrient material (e.g., nitrogen). Data may be used to forecast production, biomass accumulation and growth rate, ground cover, early vigor, senescence rate calculation, and biotic and abiotic stress identification. We used periodical NDVI values to calculate the senescence rates (SRs) and maximum NDVIs of genotypes using the portable NDVI and the process mentioned by Mullan and Mayr (2012) [23]. Plants lose their vegetative greenness at the grain-filling stage. This loss may occur due to high disease intensity, physiological disorder and plant aging, or even in the early stages. The loss of plant greenness is called senescence, and the rate of decreasing greenness over time is called the senescence rate (SR). We calculated the SR from the periodical NDVI value simply by calculating the slope of the NDVI values over time.

#### 2.1.4. Thousand-Grain Weight (TGW)

The TGW is the weight of a thousand grains in grams. To collect TGW data, ImageJ software [24] was used. For this purpose, approximately 500–1000 uncounted grains from each plot’s seed bag were spread on a white sheet with uniform light on top. The image was captured using an 18-megapixel Canon DSLR camera connected to the computer. The captured image was processed using the particle count module of ImageJ software [24].

#### 2.1.5. Grain Yield (GRYLD)

Whole plots were harvested with a Wintersteiger plot combine. All the seed bags were dried on threshing flour under hot sun for a day before weighing. The grains were weighted in an electric balance using the Android-based Inventory app [25] connected to the tablet. The weight of each plot was entered automatically from the balance to the tablet.

#### 2.1.6. Photo-Growing Degree Days (PGDDs)

Photo-growing degree days (PGDDs) combine growing degree days (GDD) and photoperiod factors on wheat development between emergence and floral initiation. Growing degree days were calculated using the method in [26], separately and in combination with the photoperiod. The equations in [26] used for the GDD calculation are as follows:(1)$$\alpha =\ln2/\ln\left(\frac{Tmax-Tmin}{Topt-Tmin}\right)$$
(2)$$Numerator=2{\left(Tav-Tmin\right)}^{\alpha }\times {\left(Topt-Tmin\right)}^{\alpha }-{\left(Tav-Tmin\right)}^{2\alpha }$$
(3)$$Denominator={\left(Topt-Tmin\right)}^{2\alpha }$$

Then, the growing degree days are as follows:(4)$$WEDD=\left(\frac{Numerator}{Denominator}\right)\left(Topt-Tmin\right)$$
where WEDD is the degree days estimated in [26]: WEDD = 0 if Tav < Tmin or Tav > Tmax. Tmin = 0, Topt = 27.7, and Tmax = 40 were the cardinal temperatures for WEDD computations before anthesis, whereas Tmin = 0, Topt = 32.75, and Tmax = 44 were the cardinal temperatures after anthesis.

According to the APSIM [27] wheat module, the photoperiod factor $\left(f_{D}\right)$ is calculated as follows:(5)$$f_{D}=1-{0.002R_{p}(20-L_{p})}^{2}$$
where $L_{p}$ is the day length in decimal format, $R_{p}$ is the sensitivities to the photoperiod that are cultivar-specific and specified by photo-sens in wheat.xml. The default value of $R_{p}$ is 3.

Finally, the PGDDs were calculated by multiplying the WEDD by the photoperiod factor, as follows:(6)$$PGDD=WEDD\times \;f_{D}$$

### 2.2. Genotyping

Genome-wide association studies (GWASs) were conducted using genotyping-by-sequencing (GBS) facilitated by next-generation sequencing (NGS) [28]. GBS is an incredibly powerful tool for swiftly detecting genome-wide polymorphisms. Because of its simplicity, resilience, repeatability, reduction in complexity in large complex genomes, and low time and cost per sample, the GBS technique is a popular choice for association-mapping investigations [29,30,31]. The GBS technique entails constructing a truncated representation of the genome, followed by adaptor ligation, pooled library PCR amplification, and multiplex sequencing. Although several target enrichment strategies are available for complexity reduction, restriction enzymes have been demonstrated to be favorable in terms of speed, specificity, and the ability to target low-copy genomic locations that sequence-capture approaches cannot reach [32]. The maize and barley GBS technique, first introduced in 2011, utilized a solitary restriction enzyme (ApeKI) and a pair of double-stranded adapters, specifically the barcode and common adapters [32]. This method was expanded to barley and wheat by using a two-enzyme system comprising a rare cutter (PstI) and frequent cutter (MspI) together with Y-adapters to build consistent libraries [30]. This method demonstrated the resilience of GBS for species with vast, complex, and polyploid genomes without a reference sequence. Every line from all three seasons was profiled using the GBS protocol of [31] and on an Illumina HISeq2500. SNP markers were identified using the TASSEL v5.2.7 pipeline [33] and mapped to the Chinese Spring Wheat Assembly v1.0 reference [34]. Genotyping calls were retrieved and filtered to ensure that the percentage of missing data per marker was less than 40%, the rate of heterozygotes was less than 10%, and the minor allele frequency was 5%. A total of 16,152 SNP markers for season 1, 16,771 SNP markers for season 2, and 12,253 SNP markers for season 3 were obtained after filtering, and missing data were imputed using Beagle v4.1 [35]. All these filtered marker data were obtained through the “Feed the Future Lab for Applied Wheat Genomics” of Kansas State University, USA.

### 2.3. Statistical Analysis

#### 2.3.1. Mixed-Effects Model Analysis for Multiple Environments

The study utilized a mixed-effects methodology through the application of Restricted Residual Maximum Likelihood/Best Linear Unbiased Prediction (REML/BLUP) analysis [36,37]. The statistical model is expressed as y=Xm+Zg+Wb+Ti+Qp+ε, where y denotes the data vector. The vector m represents the effect of the measurement–replication combination added to the overall mean, while g denotes the vector of the genetic effect. The block effect is represented by the vector b, and i represents the vector of the genotype × measurement effects. The vector p denotes the permanent environment, and ε represents the vector of the residual. The incidence matrices for these effects are represented by X,Z,W,T and Q.

The heritability was calculated utilizing the mean value, as the genotypes were replicated through the implementation of blocks. The aforementioned equation was employed to compute the heritability of the mean, where $h_{gm}^{2}=\frac{{\hat{\sigma }}_{g}^{2}}{\left[{\hat{\sigma }}_{g}^{2}+{\hat{\sigma }}_{i}^{2}/e+{\hat{\sigma }}_{e}^{2}/\left(eb\right)\right]}$. The aforementioned equation pertains to the estimation of genotypic variance (${\hat{\sigma }}_{g}^{2}$), the genotype–environment-interaction variance ${\;(\hat{\sigma }}_{i}^{2}$), and the residual variance $\;({\hat{\sigma }}_{e}^{2}$), given the number of environments and blocks (e and b, respectively).

#### 2.3.2. Mixed-Effects Model Analysis for the Single Environment

The study deployed the single-experiment mixed-effects model [38] to analyze the individual environment. This involved the use of the equation $y_{ijk}=\;m+g_{i}+r_{j}+b_{jk}+e_{ijk}$, where $y_{ijk}$ represents the response variable of the ith genotype in the kth block of the jth replicate. The intercept is denoted by m, while $g_{i}$ represents the effect for the jth genotype, $r_{j}$ represents the effect of the jth replicate, $b_{jk}$ represents the effect of the kth incomplete block of the jth replicate, and $e_{ijk}$ represents the plot error effect corresponding to $y_{ijk}$.

#### 2.3.3. Confidence Interval of Pearson’s Correlation

The study utilized a Gaussian-independent estimator for estimating the confidence interval of the Pearson’s correlation coefficient [39]. This approach was employed to mitigate imprecise estimates of the correlation coefficient and reduce multicollinearity in the multivariate analysis when examining trait associations. The width of the confidence interval was modeled through a nonlinear power model, as expressed by the following equation: $CI=\delta^{r}\times \beta_{0}\times n^{\beta_{1}}+\varepsilon$, where CI is the predicted value of the confidence interval, $\delta^{r}$ is the adjustment factor for the intercept $\left(\beta_{0}\right)$ that varies according to the strength of the relationship (r), n is the sample size, $\beta_{1}$ is the exponential rate of decay, and ε is the model residual.

#### 2.3.4. Single-Environment Multi-Trait Genotype–Ideotype Distance Index (MGIDI)

The MGIDI was computed in order to assess the simultaneous selection for the mean performance across various traits within each respective environment [40]. The computation was performed to assign a weight of 100 to the mean performance in a singular environment based on the given criteria. The ${MGIDI}_{i}$ was computed using the formula ${MGIDI}_{i}=\sqrt{\sum_{j=1}^{f}{\left(F_{ij}-F_{j}\right)}^{2}}$, where ${MGIDI}_{i}$ is the multi-trait genotype–ideotype distance index for the ith genotype; $F_{ij}$ is the score for the ith genotype in the jth factor (i=1.2…,g; j=1, 2, …,f), with g and f being the number of genotypes and factors, respectively; and Fj is the jth score of the ideotype. The genotype exhibiting a lower MGIDI is comparatively more proximate to the ideotype, thereby exhibiting desirable values for all the scrutinized traits. The degree of selection pressure can be expressed as a proportion of the complete set of genotypes through the use of selection intensity (SI) arguments, which range from 0 to 100 as integers. In this index, the term “argument ideotype” is employed to denote the trait’s superior or inferior value, which is deemed necessary for computing the ultimate index. The present model was fitted and utilized for the purpose of internally conducting heritability extraction, which enabled the automatic computation of the selection gain for the traits or factors. The anticipated genetic gain achieved through the utilization of the index SG (%) was calculated for each trait and expressed as a percentage of the selection intensity: $SG(\%)=\frac{\left({\bar{X}}_{s}-{\bar{X}}_{o}\right)\times h^{2}}{{\bar{X}}_{o}}\times 100$. The formula for calculating the mean of the selected genotypes denoted ${\bar{X}}_{s}$ involves the mean of the original population represented as ${\bar{X}}_{o}$ and the heritability coefficient denoted as $h^{2}$.

#### 2.3.5. Genome-Wide Association Mapping (GWAS)

TASSEL v5.2.7 implemented genome-wide association mapping by employing a mixed linear model [41] that takes into consideration both population structure and kinship. The first two principal components [42] accounted for population structure, whereas the pedigree–relationship matrix accounted for kinship. The mixed linear model was run using the best degree of compression and the approach of “population parameters previously defined” [43]. We utilized a Bonferroni threshold level of 0.20 to adjust for multiple testing and to find the relevant markers in each genotype group. The significant markers were then demarcated into LD-based QTLs between markers, with markers *p* < 0.001 in the same QTL for the existence of LD. The genomic locations of the markers were derived from publicly accessible mapped markers in the Triticeae Toolbox database “https://triticeaetoolbox.org (accessed on 12 June 2021)”. The trait-associated markers, as well as previously published genes or QTLs near the significant markers, were embedded against a genotype–phenotype map aligned to RefSeq v.1.1 [34].

#### 2.3.6. Statistical Software

The statistical analyses were performed utilizing R 4.0.3 software [44]. The R package Multi-Environment Trial Analysis-Metan 1.11.0 [45] was utilized to perform the analysis in sections A, B, C, and D, using the functions gamem_met(), gamem(), corr_ci(), and mgidi(), respectively. Additional data visualization and graphical representations were generated utilizing the package Tidyverse 1.3.0 [46]. The genome-wide association study (GWAS) was carried out using Tassel-v5.2.1 [47], and the map was visualized with Phenogram “http://visualization.ritchielab.org/phenograms/plot (accessed on 20 July 2021)”).

## 3. Results

Phenotypic variations, trait responses, and genotypic stability for the shifting planting times are presented in the first part of the results. Then, genomics and QTL mapping in light of the marker–trait association and GWAS are illustrated in the second part. The results are presented and discussed in detail, considering the available literature in the context of the relevant available literature, as follows.

### 3.1. Adaptation of Wheat Genotype in Early Planting by Modification of Agro-Morphological Traits

#### 3.1.1. Likelihood Ratio Test, Variance Components, and Overall Performance

The statistical analysis using the likelihood ratio test demonstrated a significant impact of the genotype on all the trait categories in the single-environment analysis. The statistical significance of the likelihood ratio for genotypes (LRTg) and the likelihood ratio for the genotype and planting time interaction (LRTge) were observed in seasons 2 and 3 for all the phenological traits, except for the LRTg in the case of BTH. Similarly, all traits in the group plant stature were significant for the LRTg and LRTge, except for the LRTge for the FLGLFW in season 1 and the SplLng in season 3. All physiological traits except the NDVI_DTB were non-significant for the LRTg in season 3. Two more non-significant likelihood ratios were reported for physiological traits in season 2: the LRTg for EGC and the LRTge for the CTIR. The TGW and grain yield were significant in the likelihood ratio test for the LRTg and LRTge in all three seasons (Table 1 and Table 2).

Each season, the average deviation was higher, showing a significantly larger genotypic response in the early-planting context (Table 3 and Table 4). Consequently, the range of the genotype selection pertaining to these traits is expanded. Most of the genotypes exhibited an increase in grain yield during the early planting time in seasons 1 and 2, while sustaining a stable grain yield across both planting times in season 3. Moreover, it was observed that the genetic factors contributed significantly to the variance in most traits across all three seasons, indicating that the genotypes had a consistent impact on the expression of traits under both planting conditions (as depicted in Figure 1, Figure 2, Figure 3 and Figure 4). Variations in both the overall performance and variance components were detected among the different trait groups.

#### 3.1.2. Adaptation for Phenological Traits in Early Planting

The phenological traits had a higher genotypic component of variation than the residual components in a single-environment analysis. Early planting had a less residual component of variation than the timely-planting conditions for all the phenological traits in a single-environment analysis (Figure 1). Significant genotype–environment interaction (GEI) was also observed for all the phenological traits. The higher GEI values seen in the phenological traits are evidence that the GEI influenced the phenotypic variances in these traits. The analysis indicates that during the first and second seasons, the DAYSMT exhibited limited GEI, suggesting that genotypes played a dominant role in determining the number of days required for full maturity. However, GEI had a discernible effect on other traits, such as the BTH and GFD (Figure 2). Phenological traits for calendar days and photo-growing degree days were found to differ in both the overall performance and variance components. The study observed phenological traits with respect to calendar days and found that genotypes exhibited varying levels of aptitude for DTHD responses during early sowing in comparison to other traits over the course of three years. Interestingly, when we incorporated photoperiods and temperature to calculate the photo-growing degree days (PHDDs), the variable aptitude in season 3 was minimized, and a straightforward consequence of longer PHDDs was found in the early-planting conditions rather than in the timely-planting conditions for each of the seasons.

Early planting resulted in distinct responses for the remaining phenological traits. Under early-sown conditions, genotypes exhibited a prolonged duration in the booting stage compared to those planted in a timely manner. The longer PG_DTHD and PG_GFD were the consequence of longer PG_DAYSMT for each of the seasons. We employed PHDDs for future investigation instead of straightforward calendar days for phenological features because it is more precise and utilizes environmental variability for phenological expression. Several prior studies also supported the use of the photoperiod and rising degree days [48,49,50].

#### 3.1.3. Adaptation for Plant Stature in Early Planting

Early planting resulted in a significant increase in the plant height, with the exception of season 2, as illustrated in Figure 2. The reason for this phenomenon can be attributed to the occurrence of hailstorms in the Ludhiana region of India. The occurrence of a hailstorm during the heading stage may have played a role in the reduction in the plant height, particularly impacting the length of the peduncle and spike. The genotypes exhibited a reduction in the flag-leaf width, while an increase in the flag-leaf length was observed during early planting. As a result, the flag-leaf area exhibited no significant variation across the two planting periods. The height up to spike (HUS) does not seem to be different from the PH, and we used it for further progress in our study, as it excluded the spike length (SpkLng), which we considered separately in our study. The peduncle length (PDL) and the length from the ground to the peduncle node (PDG) were considered for the plant stature study for season 2 and season 3. Unfortunately, the study of the PDG and PDL did not seem consistent in either year; that is why we discarded these traits from further study. Lower GEI in the plant height was observed in both seasons 1 and 2, but it was found to be more pronounced in season 3. Henceforth, conducting targeted research that centers on the interpretation of genotypic heterogeneity through a distinct assemblage of genotypes could prove advantageous.

#### 3.1.4. Adaptation for Physiological Traits in Early Planting

The mean deviation observed for the EGC during early sowing was comparatively low in contrast to planting at the appropriate time (Table 3 and Table 4).

Higher ground cover was observed in early planting than in timely planting, except in season 3. The NDVI value at booting and heading was higher in early planting than timely planting, except in season 2. Hailstorms during booting to heading in season 2 caused the lower NDVI. The maximum NDVI value was also higher in seasons 1 and 3 for early planting than for timely planting. Outliers below the 25th percentile for the NDVI indicated that few genotypes had very low greenness due to poor stand establishment in both the planting times. Timely planting had a higher senescence rate (SR) and canopy temperature increasing rate (CTIR) in all three seasons. All the physiological traits had higher residual components of variations compared to the other trait groups.

#### 3.1.5. Adaptation for TGW and Grain Yield in Early Planting

A higher TGW was found in the early- rather than the timely-planting condition for seasons 2 and 3, whereas there was no significant TGW difference for season 1 in the shifting planting times. The residual variance component of the GEI for the TGW in season 1 was also less compared to the other two seasons. The GEI was observed to be higher in grain yield owing to its dependence on both inherited genetic factors and environmental conditions, as well as its high variability as a trait (Figure 4). The extended duration of winter during season 3 was observed to be the probable cause for the uniform grain production at the two planting times during year 3, as depicted in Figure 4a.

#### 3.1.6. Multi-Trait Stability Index, Ideotype Design

The study found that phenological events exhibited a heightened impact on both the grain yield and thousand-grain weight (TGW). In the analysis of the PG_DTB, PG_BTH, and PG_GFD, it was observed that longer PG_DTB exhibited a robust positive correlation with the GRYLD and TGW, thereby indicating its potential to enhance the yield and TGW (Appendix A). The study indicates that an extended vegetative period is anticipated in the case of early planting, as evidenced by the overall performance analysis. To promote a prolonged vegetative phase during early planting as opposed to timely planting, we applied selective pressure to encourage an extended PG_BTH during early planting and a shortened PG_BTH during timely planting.

It is anticipated that a prolonged period of grain filling (PG_GFD) will facilitate an increase in both the grain yield and grain weight. The prompt initiation of planting may expedite the plant’s maturation process because of the manifestation of terminal heat stress while it undergoes the grain-filling stage. The findings of the study indicate that an increase in plant height is positively correlated with a higher grain yield and thousand-grain weight (TGW). The ideotype was favored to have a reduced height to induce selection pressure for lodging tolerance, as indicated in Table 5. The aforementioned determination has been corroborated via various additional investigations [51,52]. In addition, a study conducted in Australia discovered a significant lodging occurrence in wheat plants during early planting [53].

Rapid and robust ground coverage is a desirable outcome for optimal crop establishment in all planting scenarios. The desire for robust grain filling and, subsequently, higher yields necessitates the delay of senescence and the slowing of the canopy temperature. Given the positive correlation between a higher TGW and an increased grain yield (Appendix A), we incorporated this trait into the ideotype design. Table 5. displays the ideotype design utilizing the MGIDI in its entirety. The trait combination pointed out within the scope of this study possesses the potential to incite further investigation endeavors focused on the alteration of wheat plants [54].

Except for a few traits that did not match the selection gain, the MGIDI’s prediction of the genetic gain for each trait shows that the logic used to create the ideotype was sound. The study recorded a significant increase in the PG_DTB for the TP and in the PG_BTH for the EP across all planting conditions, including the FLGLFL, FLGLFW, and GRYLD. In addition, the intended adverse outcome was observed in both the SR and CTIR under the planting conditions across all three seasons, apart from the TP in S1 (Table 5).

The selection pressure exerted on a limited number of traits was found to be incongruous with the intended selection gain. The phenological traits, namely, PG_DTB for early planting in S1 and S2, PG_BTH for timely planting in S2 and S3, and PG_GFD in all seasons, were found to be incongruent with the intended selection pressure. The results indicate that there was a lack of consistency between the EGC for early planting in all three seasons and timely planting in S3. The TGW exhibited a lack of conformity during the initial planting phase, while in S3, it failed to align with the intended selection pressure during the timely-planting phase. Ultimately, there was a lack of synchronization between the HUS and timely planting in S2 and S3, as well as in S3 during early planting. The observed discrepancy can be attributed to the genotypic capacity to execute particular traits. The incongruent selection pressure in phenology indicated that a longer vegetative and reproductive period is always supportive of higher yields. The mismatch in the selection gain for the TGW in early planting indicated that an increased TGW might not be good enough for genotype selection in early establishment. A higher EGC did not support selection gain for early planting. A high level of strength in these factors indicates that the traits encompassed by the factors exerted significant influence on the chosen genotypes, which will enable subsequent breeding initiatives to select genotypes on the basis of this factorial analysis. The breeder must possess a high level of discernment in identifying trait-based specific genotypes to facilitate early establishment within the chosen genotypes, as depicted in Figure 5. The MGIDI-based analysis of the genotypic performance revealed that eight genotypes in S1 and S2 and three genotypes in S3 exhibited favorable performances under both planting conditions, as depicted in Figure 6. The MDIGI was utilized to identify the most optimal genotypes in each year, taking into account both sowing conditions.

### 3.2. Identification of QTLs Associated with Adaptation to Early Planting

#### 3.2.1. Population Structure

The highest number of markers were present in the B-genome (48.7% in S1, 50.0% in S2, and 40.3% in S3), followed by the A-genome (39.2% in S1, 37.8% in S2, and 39.7% in S3) and D-genome (10.6% in S1, 10.7% in S2, and 17.8% S3). The population structures of the three seasons’ genotypes were studied to assess the genetic diversity of the wheat genotypes in the experiment. The highly annotated and potentially large phylogenetic tree from the filtered GBS data of each season was generated using Archaeopteryx in TASSEL 5, which elucidated that the genotypes used in the study were highly diverse in their genetic architecture and can generate polymorphic markers for GWAS analysis for traits of interest (Figure 7).

#### 3.2.2. Genome-Wide Association Mapping for Multiple Traits in EP and TP

The genome-wide association was performed using multiple-traits data obtained from the phenotypic study of objective-1. After the Bonferroni correction for the multiple testing, we obtained 2347 markers significantly associated with our traits of interest. The significant markers for the grain yield are elucidated in the Manhattan plot of the MTA in Figure 8. The chromosomes are shown in the *x*-axis and the −log10 *p* values in the *y*-axis. The threshold values for each of the traits concerning seasons and planting times align with the threshold values obtained through the implementation of the Bonferroni correction for multiple testing at significance levels of 0.01 and 0.20, respectively. All the Manhattan plots can be found in Appendix A. The significant markers, mapped in the IWGSC Ref map 1.1, are presented in Appendix A.

It was found that a higher number of significant markers was found in early planting (1036) compared to timely planting (819). Chromosome 5B was the highest source of significant markers, followed by Chromosome 2B for early planting (Appendix A). Similarly, when we observed the number of significant markers for specific traits in the different planting conditions across the season, we found the highest number of SNP markers for the PG_GFD (186), followed by the PG_DTB (140) and FLGLFL (102) in early planting, whereas the highest number of significant markers were found for the FLGLFL (132), followed by the SR (103) and PG_BTH (99).

##### QTLs Associated with Morphological Traits

LD analysis was performed in TASSEL by selecting markers R2 above 0.75 and D’ above 0.85 to identify the QTLs associated with the traits of interest. After LD analysis, we identified a total of 96 unique QTLs. Among them, 33 QTLs were associated with multiple traits in different planting times (Appendix A). A total of 44 of them were found to be associated with early-planting traits, and 31 were found to be associated with timely-planting conditions. Additionally, we found twenty-two QTLs that were associated with both planting times. The highest number of QTLs were found in chromosome 7A for both the planting times, whereas chromosomes 4A, 4B, and 4D showed smaller numbers of associated QTLs (Appendix A).

##### QTLs Identified Related to Phenology-Affecting Genes

Mostly, wheat phenology can be affected by the photoperiod, vernalization, and regional adaptation to modify the wheat life cycle. The genes that control the vernalization requirements, photoperiod sensitivity, and even the epistatic effect of earliness per se to modulate the life cycle for regional adaptation could have extended effects for early adaptation [55]. The current study shows that several QTLs are linked with these genes (Figure 9 and Table 6).

Among the photoperiod-sensitive genes, *Ppd-B1* is located 2.86 Mbps distant from the QTL QMpt.bisa.2B.2 on chromosome 2B, which is associated with the senescence rate and photo-growing degree days from the booting to heading days in early-planting conditions. Interestingly, the *Ppd-D1* allele was found inside the QTL QMpt.bisa.2D.3 on chromosome 2D, which is associated with early ground cover, photo-growing degree days, booting to heading days, and the grain-filling period in the early-planting condition, whereas this QTL is associated with the flag-leaf width in the timely-planting condition. Among the vernalization alleles, *Vrn-A1*, *Vrn-A3*, *Vrn*-*B1*, and *Vrn-D3* were found to be near enough some associated QTLs. The QMpt.bisa.5A.3 is 5.68 Mbps distant from the *Vrn-A1* allele. This QTL is associated with multiple traits, such as the EGC, FLGLFL, HUS, PG_GFD, and TGW in early-planting conditions, and with the traits PG_BTH and FLGLFL in timely-planting conditions. The QTL QMpt.bisa.7A.4 is very close to the *Vrn-A3* (0.08 Mbps) allele, which is associated with the GRYLD in early planting and with the PG_BTB in timely planting. *Vrn-B1* on chromosome 5B is close to two QTLs: QMpt.bisa.5B.3 (2.54 Mbps) and QMpt.bisa.5B.4 (9.02 Mbps), which are associated with the SR and PG_DTB in early-planting conditions. The QTL QMpt.bisa.5B.4 is also associated with the GRYLD in early-planting conditions. Finally, the QTL QDtb.bisa.7D.3 on 7D is 0.13 Mbps distant from *Vrn-D3.* This QTL is associated with the PG_DTB in timely-planting conditions. The other *Vrn* alleles were farther distant (from 61.16 to 262.19 Mbps) from the identified QTLs. The QTLs QBth.bisa.1D.4, Qhus.bisa.1D.1, and QMpt.bisa.1D.3 are distantly located from the *Eps* loci at chromosome 1D, which suggests that the identified QTLs are noble and may not be associated with *Eps* loci. Further research is needed to identify an exact MTA analysis for these noble QTLs.

##### Trait-Attributing QTLs and Genomic Regions in Different Planting Times

The 96 QTLs were found to be attributed to different traits for the early- and timely-planting conditions. Some of these QTLs were common for both planting times. The effect of these QTLs on agronomic traits may be taken into consideration with further fine mapping for early adaptation behaviors.

##### Phenology-Associated Genomic Region

Fifteen QTLs were significantly associated with the PG_DTB for early planting, and six QTLs were associated for timely planting with the same trait in different chromosomes. Among them, two QTLs, *QDtb.bisa.2D.4* and *QMpt.bisa.5B.2*, were found common for both planting times. Thirteen QTLs were significantly associated with the early planting time, whereas ten QTLs were significantly associated with the timely-planting conditions for the PG_BTH. All those QTLs for the PG_BTH were different for early and timely planting, which suggests adaptation for planting time control by different QTLs. Among the four QTLs for the PG_GFD in the timely-planting conditions, three were common for the early-planting conditions. However, a large set of 21 QTLs in different genomes was significantly associated with the PG_GFD in the early planting time (Table 7). These increased QTLs in early planting for the PG_GFD suggest that more genomic regions with different genes might influence the early adaptation.

##### Plant-Stature-Associated Genomic Region

The plant height at different planting times was controlled by different QTLs, except the QTL *QMpt.bisa.7A.6*. Seven unique QTLs for the early-planting condition and six unique QTLs for the timely-planting condition were significantly associated with the HUS. Four common QTLs were significantly associated with the FLGLFL, whereas nine unique QTLs for the early-planting condition and twelve unique QTLs for the timely-planting condition were found to be significant with the same trait. Seven QTLs were found to be significantly associated with the FLGLFW in timely planting, and four QTLs for the early-planting condition, along with one common QTL: *QMpt.bisa.6A.3* (Table 7).

##### Physiology-Associated Genomic Region

Among the physiological traits, a higher number of QTLs were found to be significantly associated with the SR for both planting times. Eleven unique QTLs in early planting and six unique QTLs in timely planting were considerably associated with the SR. Most of these QTLs control multiple traits. For the QTLs that control EGC, seven unique QTLs in early planting and two in timely planting were found to be significant. Finally, no significant QTLs were found to control the CTIR in early planting, but six QTLs were significantly associated with this trait in the timely-planting condition (Table 7).

##### TGW–Yield-Associated Genomic Region

The TGW and GRYLD are highly quantitative traits that depend on several molecular activities. There were nine unique QTLs found to be significantly associated with the TGW in early planting, and four unique QTLs found to be significantly associated with the timely-planting conditions for the TGW. Eight QTLs were significantly associated with the GRYLD in early planting, and four in the timely-planting conditions. Among these yield-associated QTLs, one QTL, *QMpt.bisa.3A.4*, was common for both planting times (Table 7).

## 4. Discussion

Seasonal variations often influence adaptation by changing the temperature and photoperiod. Three different sets of genotypes were grown in different seasons, with varying weather conditions. The photo-growing degree days were measured to standardize the temperature and photoperiod with calendar days. It was evident that the photo-growing degree days reduced the data variability by minimizing the calendar-day difference with the PGDD calculation in phenological traits [48]. Calendar days were inconsistent for the DTB and DTHD, but it was clearly evident that longer PGDDs are required more in early planting than timely planting. In season 2 (S2) and season 3 (S3), most genotypes exhibited longer days to booting (DTB) and days to heading DTHD when planted timely manner (TP). However, upon incorporating the photo growing degree days (PGDD) to calculate PG_DTB and PG_DTHD, the genotypes demonstrated shorter PGDDs values in both seasons, as compared to early planting (EP). The seasonal ambiguities were high in both these two seasons. Hailstorms along with high rainfall occurred in S2 during the booting and heading time of the timely-planted crops.

Similarly, prolonged winter in S3 caused the longer DTB and DTHD for the TP. Lower temperatures exist in both these seasons, increasing the phenological period. The addition of the cumulative effect of temperature and the photoperiod by the PGDDs minimized these seasonal ambiguities. It has been suggested that the growth and development of the crop is a function of the photoperiod and ambient temperature [66,67].

The global wheat-breeding program implemented by CIMMYT has demonstrated consistent genetic improvement in grain yields across diverse environments and management conditions worldwide, with a particular emphasis on South Asia [68]. While certain traits have exhibited consistent improvement over time. Unique methodologies have been employed with the established cultivars to attain maximum grain productivity [69]. According to a study, a considerable proportion of developing countries’ spring-wheat-growing regions (approximately 70%) either utilize CIMMYT germplasm as a parent of their varieties or produce CIMMYT germplasm as an immediate release [68]. This practice has resulted in noteworthy economic advantages. The genotypes utilized in this investigation were sourced from an extensive collection of advanced lines produced within the wheat-breeding program of CIMMYT. The genotypes were specifically identified for conditions that correspond to the early-sowing conditions prevalent on India’s >20-million-hectare Indo-Gangetic plain, which are referred to as ME1 and ME4 environments.

A number of yield-contributing factors have been identified as suitable for early planting, which can mitigate yield reductions resulting from increasing seasonal temperatures by achieving maturity well before the onset of terminal heat stress [1]. Nevertheless, initiating the sowing process at an early stage is accompanied by the potential hazard of inadequate initial establishment and accelerated growth during the initial phases owing to elevated temperatures, which may ultimately lead to diminished yields as a result of decreased dry-matter accumulation.

According to data from the National Informatics Centre under the Government of India in 2016, it has been observed that India’s seasonal air temperature has undergone changes over the past twenty years [70]. This has resulted in minor heat stress on crops during early planting. In contrast to planting at an optimal time, early planting results in a prolonged exposure of genotypes to a shorter photoperiod during the pre-flowering stage.

The findings indicate that there is no discernible variance from a previous report [71], which presumes that a decrease in the photoperiod duration is positively correlated with an increase in the duration of the growth stages in wheat. Therefore, if planting is carried out much earlier than the optimal time, then genotypes tend to remain in the booting stage for a longer duration, resulting in the complete manifestation of their genetic potential. 

According to the findings of this research, the act of planting crops earlier than usual does not necessarily guarantee an early maturation of the crop. The wheat-growing period was prolonged through early planting, primarily by extending the duration of vegetative growth. The extension of the vegetative phase, as indicated by the PG_DTB and PG_BTH, through early planting, led to a notable boost in crop yields across various genotypes. The findings indicate a noteworthy positive correlation between the TGW and PG_GFD, while exhibiting a negative correlation with other phenological traits in both planting periods. This suggests that the observed increase in the yield during early planting may be attributed to a rise in the number of grains, which is facilitated by a higher count of fertile spikelets in the spike. Increasing the duration of the vegetative stage and expanding the leaf area during early planting can lead to an increase in biomass. This increase in biomass results in the accumulation of more dry matter at the source, which is subsequently transported to the sink. Consequently, it seems to enhance both the quantity and quality of grains. According to a study, a graph estimation model indicated that the most resilient network of traits for enhancing grain yields in all environments comprised the biomass, thousand-grain weight (TGW), and grain number [69].

The ideotype design approach aims to enhance crop productivity by concurrently considering the selection of genotypes based on multiple traits. The design of an ideotype, as proposed by Olivoto and Nardino in 2020, involves the favorable selection of multiple traits with satisfactory gains for their application in breeding programs [40]. The study revealed a satisfactory genetic gain for certain traits, suggesting that the attainment of an ideotype design is feasible through the utilization of the MGIDI, whereby traits are designated for increase or decrease. A study on the selection of stress-resistant maize hybrids generated a comparable and evident conclusion to assure long-term gains in primary traits while maintaining genetic gains in secondary traits [72]. The MGIDI exhibits a superior performance in comparison to other linear selection indices, thereby aiding breeders in the identification of superior genotypes. The utilization of multiple traits for ideotype design holds the potential to mitigate the adverse effects of multicollinearity. As a result, it leads to enhanced conditioned matrices and unbiased index coefficients, which facilitate the estimation of the genetic gain [40]. The utilization of a graphical and simplified method for evaluating the strengths and weaknesses of genotypes and traits facilitated the identification of suitable candidates for continued implementation in the current breeding program.

Certain traits that do not align with the ideotype and may hinder the optimal performance of genotypes should be taken into account when planning future breeding programs. Crop breeders are required to select genotypes from a pool of advanced lines that exhibit the desired level of strength for the trait of interest in a crop-breeding program. The present investigation revealed a distinct trend of strengths and weaknesses for the traits in question, as evidenced by the prompt and punctual plantings observed across the three-year period. Phenological traits such as the PG_DTB and PG_GFD demonstrated significant selection gain throughout all seasons in early planting. However, weaker support for the selected genotypes in early planting for two seasons was observed in the TGW and GRYLD. Nevertheless, the superior results achieved by well-executed genotypes on both dates of planting exhibited greater stability compared to other genotypes. The possibility exists for their selection in future variety releases within a broader spectrum of the 20 MHA Indo-Gangetic regions, which encompass a range of climatic conditions, from cooler and drier (NWPZ) to warmer and more humid (NEPZ). The vast expanse of wheat cultivation in India constitutes a unique geographical region characterized by predominantly small–marginal farmers who seek a cultivar capable of adapting to diverse sowing periods. The cultivation of the said genotypes can be carried out either during the early sowing period or within the appropriate time frame, contingent upon the cropping system, while maintaining optimal performance.

Effective breeding programs require excellent genetic resources with a diversified gene pool in the breeding material [73]. A huge genetic variation exists across the wheat genome, and it was noticeably high for the A and B genomes in CIMMYT germplasms [56]. It was also noted in the same article that CIMMYT’s synthetic wheat germplasm has lower gene diversity in the D genome compared to the A and B genomes. The materials used in the current study also had less genomic diversity in the D genome. Again, the proper understanding of and acquaintance with the linkage disequilibrium extent are necessary to determine the adequate requirement of the marker density for proper MTA study, where it is notably true that the LD declines swiftly with distance [74]. It was observed in a recent study [65] that the marker number is no longer a critical limiting factor for prediction accuracies in wheat in which the LD is high. Moreover, counterfeit MTA might cause less diverse population structures in GWASs [75,76,77]. In this study, the population was highly stratified, giving a broadened general overview and knowledge of the biological background of the mapping panel.

The GWAS revealed several significant marker–trait associations. The marker diversity in the D genome was lower compared to the A and B genomes. A similar result was found in the genetic analysis of the spring wheat association-mapping panel [78]. The GWAS for the trait of interest in early adaptation was found to be valuable in the genetic architecture study and co-localization of loci for the traits. Several captivating co-localizations were identified in a study for phenology, plant stature, disease resistance, and even yield, contributing traits with potential associations with the GRYLD and stability, and representing its possible significance in global wheat-breeding strategies [65]. Our current study also found several genomic regions with associated significant markers in several genomes for multiple traits. LD analysis removed all the false-positive markers in the GWAS and, finally, 96 unique *QTL*s were identified. Several of these *QTL*s were co-localized with multiple traits at different planting times. The co-localization of SNPs and *QTL*s for different traits in different environments has been reported in several studies [62,79,80,81,82,83,84].

The early heat tolerance can be examined by applying selection pressure to a diverse genetic materials cultivated under October sowing conditions [85]. This selection intensity is influenced mainly by several phenology-affecting QTLs or genes. In our study, we found QTLs not only associated with phenology, but also with other groups of traits, such as plant stature and physiological traits. Several heat-tolerant QTLs have been receiving increased attention over the last few years [86,87,88,89]. A review on recent GWASs on wheat over the last decade (from 2010 to 2020) found thousands of MTAs conferring abiotic stresses [90]. Among these MTAs, seedling heat tolerance was reported by Maulana et al. (2018) [87], who identified several QTLs containing potential sources of candidate genes of early heat tolerance in winter wheat. In our study, we found 44 unique QTLs associated with early adaptation in spring wheat. Among these QTLs, co-localization for various traits were common. The QTL prefix with “QMpt” in our study refers to the co-localized loci for multiple traits. Again, some QTLs were found to be co-localized even for both planting times. These QTLs are obviously crucial for definitive study, but we considered unique QTLs for different planting times. Several QTLs linked to yield component traits have been detected in the last decades through GWASs [90]; some of them were validated using bi-parental mapping population analysis [90,91] or meta-QTL analysis [92]. In early planting conditions, two unique QTLs (QMpt.bisa.2B.2 and QMpt.bisa.2D.3) linked to photoperiod-sensitivity genes, such as *Ppd-B1* and *Ppd-D1*, controlled the PG_BTH. This phenomenon confirms the importance of photoperiod sensitivity in early adaptation. Again, *Vrn-B1* was found to be linked with two QTLs (QMpt.bisa.5B.3 and QMpt.bisa.5B.4) to control the trait PG_DTB in early planting. The linkage of the days to booting with *Vrn-B1*-linked QTLs suggests a vernalization requirement in the early-adapted genotype. We can conclude a mild vernalization requirement for early adaptation, as we worked with spring wheat. Genes for photoperiod sensitivity and the vernalization requirements were also shown to make an effective contribution to early adaptation [55]. In early planting, a higher number of QTLs responsible for all three phenology-affecting traits reveals that early adaptation influences many putative genes or QTLs to exert their impacts compared to timely planting. Similar cases also were found in physiological traits, except for the CTIR. The highly quantitative traits TGW and GRYLD were also revealed to be associated with a higher number of QTLs for early planting compared to timely planting. In a nutshell, early planting is influenced by more QTLs for phenology, physiology, the TGW, and the GRYLD. The genotype–phenotype map connecting key markers to the RefSeq illustrates the RefSeq’s use as a platform for comparing and confirming GWAS results. Targeted selection for the desired region is now easy using this community resource for wheat breeding. Our study provides an opportunity for accelerating GWAS-assisted wheat breeding for early adaptation in the Indo-Gangetic region.

## 5. Conclusions

The act of planting at an early stage was observed to result in alterations in the manifestations of specific features in various genotypes. This could potentially be attributed to the impact of early heat exposure and the presence of a significantly prolonged phenological phase. The temporal placement of planting exerted a greater influence on phenological traits in comparison to other traits. The observed increase in the grain yield during early planting can be attributed to the extended phenological period, which facilitated greater mobilization from source to sink. The development of an ideotype design for subsequent breeding programs is predicated on a single-environment approach. The MGIDI was employed to identify genotypes based on their traits during both planting periods. The significance of phenological traits in achieving selection gain during early planting was established. In the context of early planting, genotypes exhibiting favorable phenological traits (namely, an extended vegetative phase and prolonged grain-filling period) were identified as effective means of achieving a higher grain yield and thousand-grain weight. A total of 44 novel QTLs and 22 common QTLs were found to be associated with early adaptation in wheat, which could be used for genotype identification in early planting. The proposition is made that the assessment of cultivars under conditions of timely and early sowing may facilitate their development with broad adaptability. However, due to the prevalence of the semi-dwarf stature within the CIMMYT germplasm, the application of selection pressure towards a reduced stature did not yield favorable results for genotype selection.

## Figures and Tables

**Figure 1 genes-14-01507-f001:**
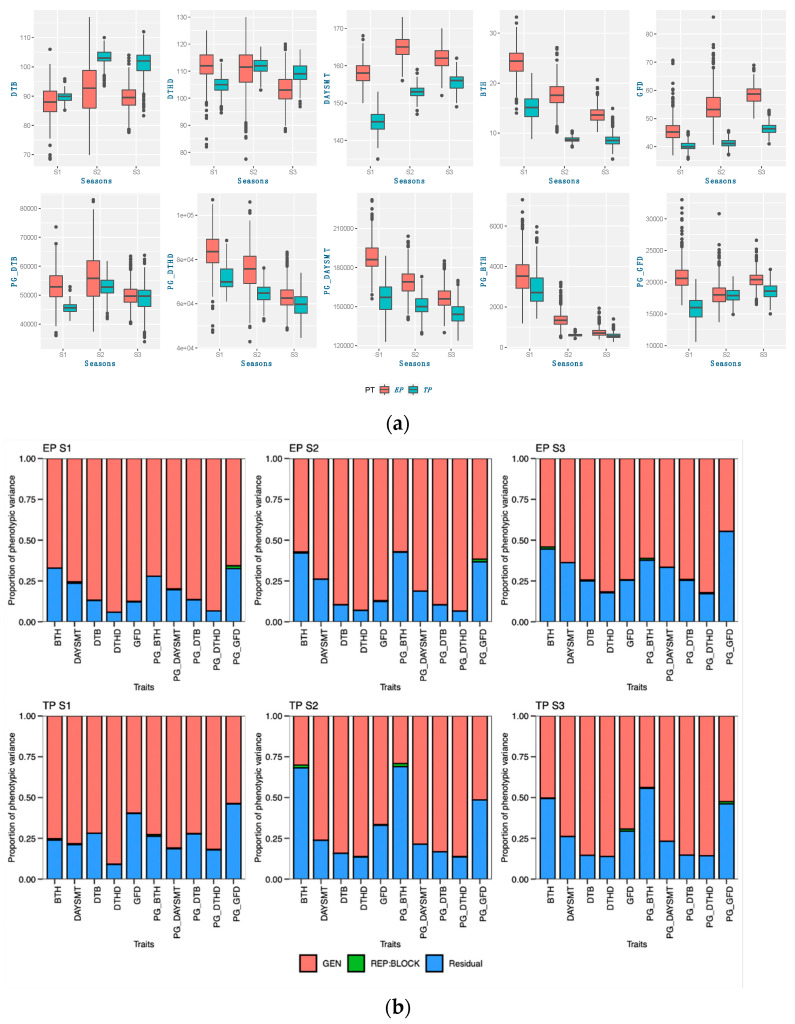
Trait properties and their variance components of phenological traits: (**a**) boxplots comparing the traits in three seasons; (**b**) variance components of single-environment analysis; (**c**) variance components of MET analysis.

**Figure 2 genes-14-01507-f002:**
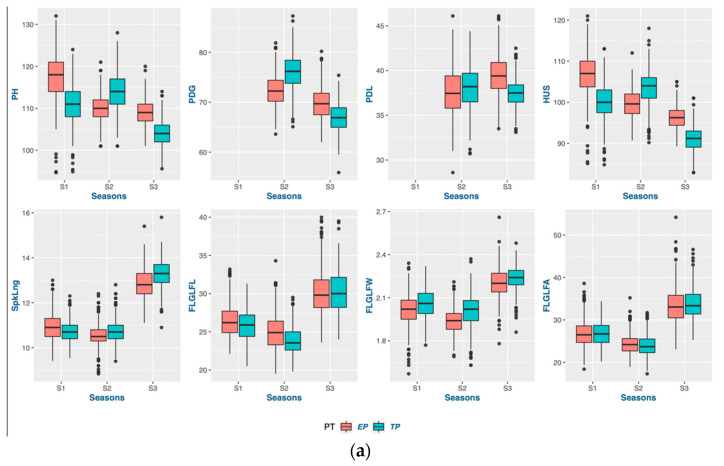
Trait properties and their variations in plant stature: (**a**) boxplots comparing the traits in three seasons; (**b**) variance components of single-environment analysis for plant stature; (**c**) variance components of MET analysis for plant stature.

**Figure 3 genes-14-01507-f003:**
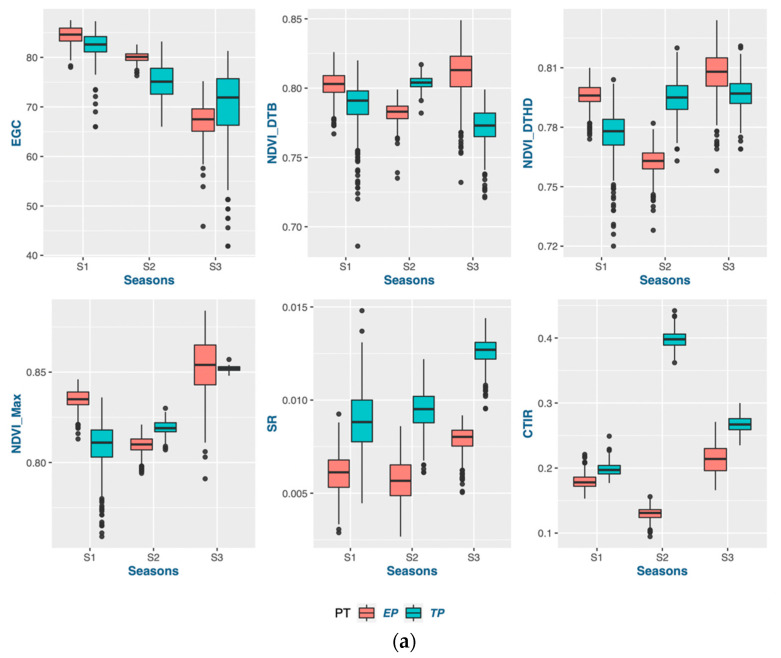
Trait properties and their variance components in physiological traits: (**a**) boxplots comparing the traits in three seasons; (**b**) variance components of single-environment analysis; (**c**) variance components of MET analysis.

**Figure 4 genes-14-01507-f004:**
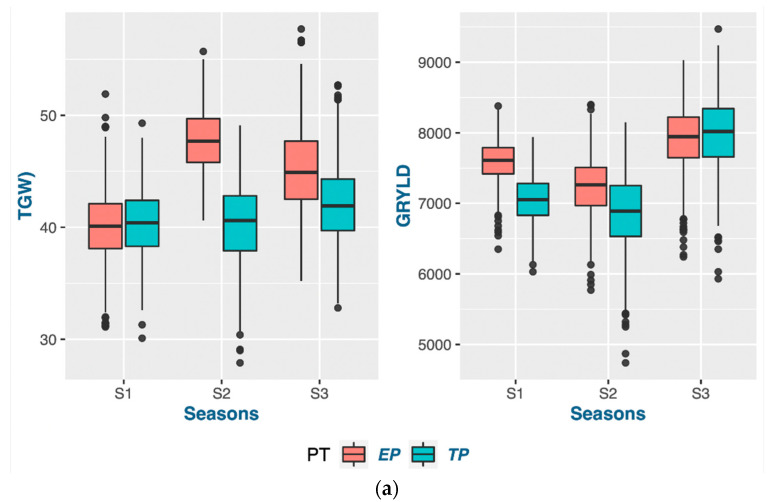
Trait properties and their variance components in grain yield and TGW: (**a**) boxplots of the variations in three seasons; (**b**) variance components of single-environment analysis; (**c**) variance components of MET analysis.

**Figure 5 genes-14-01507-f005:**
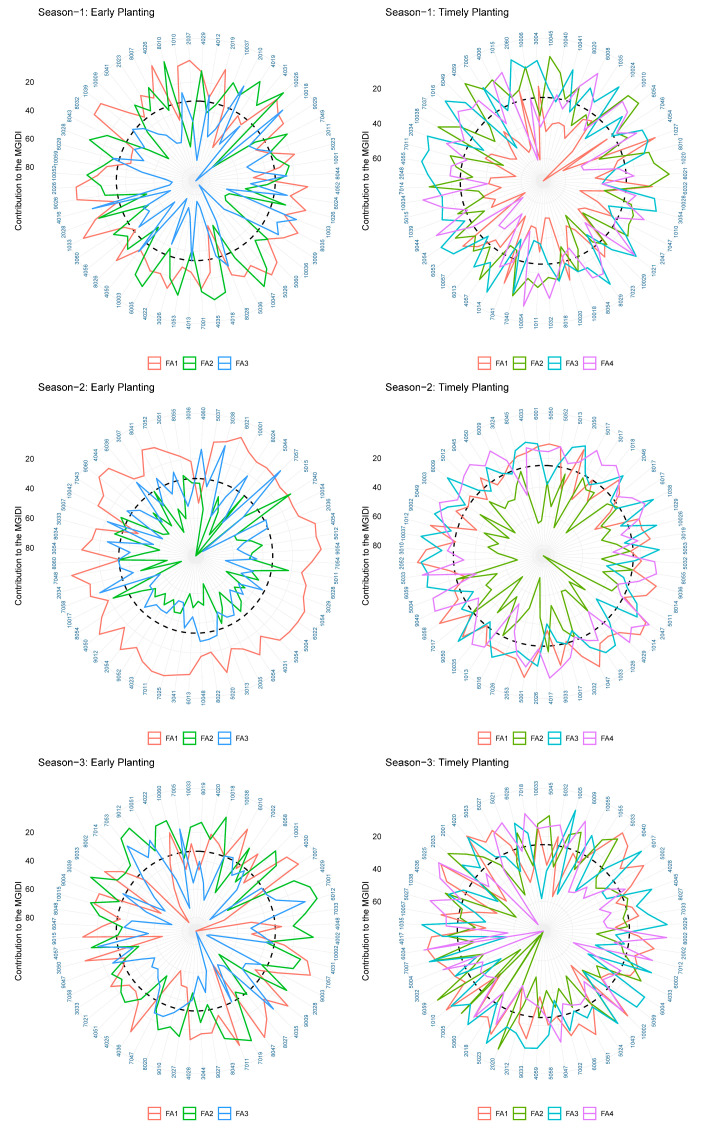
The present study investigates the MGIDI in relation to selected genotypes and factors, with a focus on identifying strengths and weaknesses. Traits that were closely related were categorized together within a single factor, while traits that were distantly related were categorized into separate factors. The factors located in proximity to the edge exhibit a reduced degree of distance from the intended ideotype, thereby engendering the selection of genotypes.

**Figure 6 genes-14-01507-f006:**
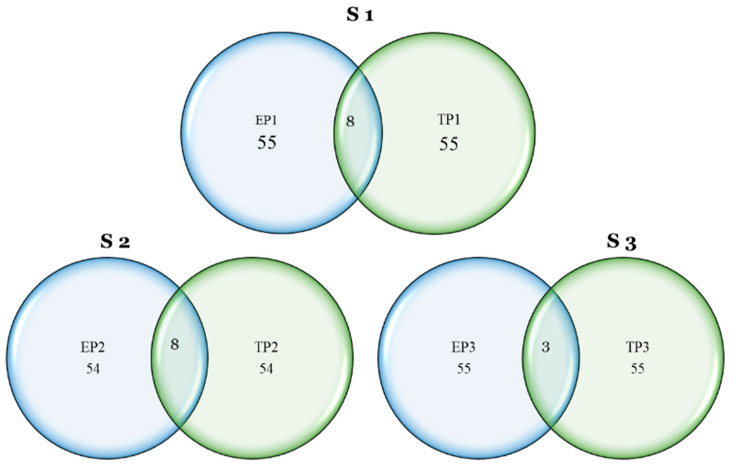
The results of the Venn diagram analysis, which focused on selected genotypes for early planting and timely planting, indicated that a total of eight genotypes performed well in both seasons 1 and 2, while three genotypes performed well in season 3 for both planting times.

**Figure 7 genes-14-01507-f007:**
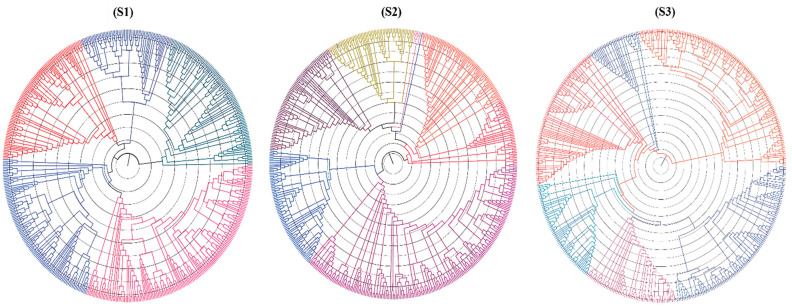
The study investigated the population structures of genotypes from GBS data and found that they have a significant amount of diversity.

**Figure 8 genes-14-01507-f008:**
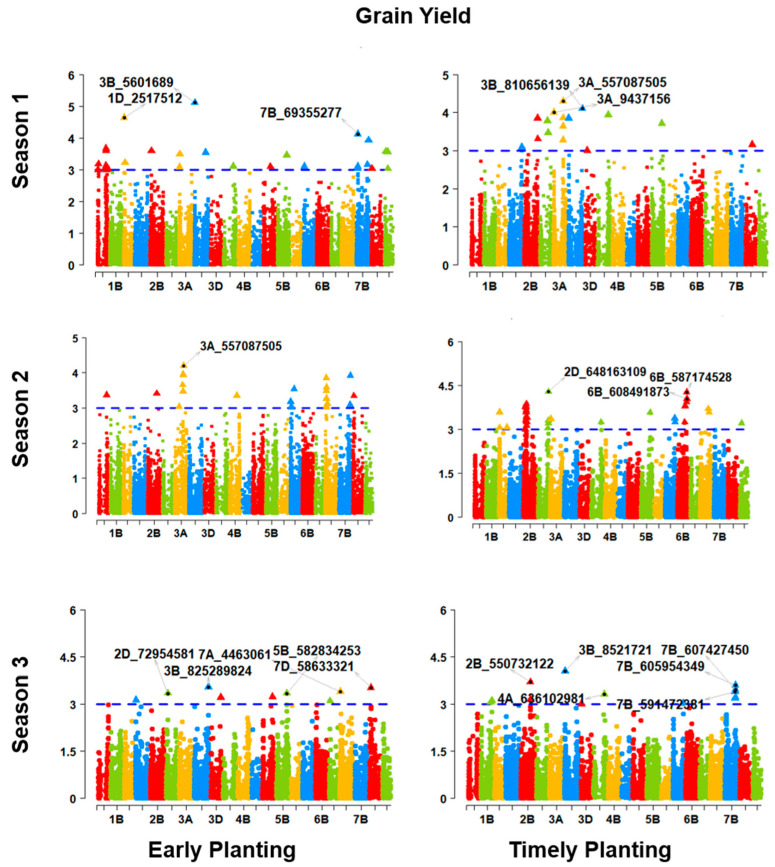
The utilization of Manhattan plots led to the identification of a number of markers that exhibited a significant association with the grain yield across the S1, S2, and S3 stages of development for both early and timely planting. Appendix A contains all additional Manhattan plots for multiple traits of interest.

**Figure 9 genes-14-01507-f009:**
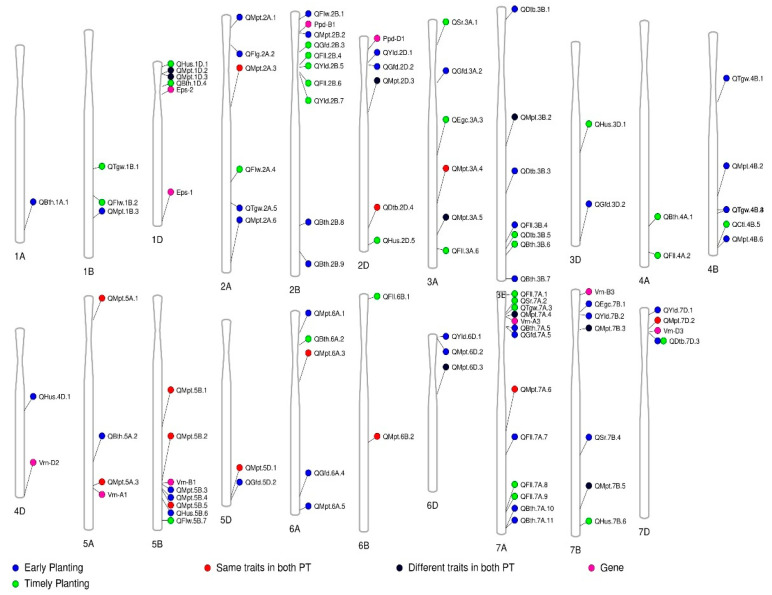
The distribution of quantitative trait loci (QTLs) across the genome, in conjunction with the presence of established phenology-affecting genes, such as *Vrn*, *Ppd*, and *Eps*, was examined for varying planting times.

**Table 1 genes-14-01507-t001:** The likelihood ratio test (LRT) was employed to examine the significance of the model in the research study. The use of the LRT for assessing the genotypic effect is a valuable tool for determining the statistical significance of traits in single-environment analysis.

Traits	Early Planting			Timely Planting		
	Season 1	Season 2	Season 3	Season 1	Season 2	Season 3
Phenology						
DTB	963 ***	1090 ***	534 ***	467 ***	791 ***	862 ***
DTHD	1420 ***	1340 ***	720 ***	1140 ***	886 ***	894 ***
DAYSMT	554 ***	506 ***	343 ***	619 ***	579 ***	506 ***
BTH	416 ***	268 ***	224 ***	559 ***	67.5 ***	183 ***
GFD	954 ***	1020 ***	523 ***	268 ***	412 ***	469 ***
PG_DTB	946 ***	1090 ***	525 ***	467 ***	761 ***	860 ***
PG_DTHD	1350 ***	1380 ***	736 ***	736 ***	884 ***	874 ***
PG_DAYSMT	667 ***	688 ***	384 ***	693 ***	635 ***	573 ***
PG_BTH	496 ***	262 ***	300 ***	511 ***	61.6 ***	129 ***
PG_GFD	374 ***	466 ***	157 ***	233 ***	259 ***	241 ***
Plant Stature						
PH	537 ***	306 ***	89.1 ***	534 ***	488 ***	119 ***
SpkLng	262 ***	75.4 ***	77.9 ***	187 ***	172 ***	51.9 ***
PDG †	-	220 ***	87.4 ***	-	332 ***	114 ***
PDL †	-	317 ***	243 ***	-	516 ***	107 ***
HUS	528 ***	329 ***	56.1 ***	536 ***	493 ***	100 ***
FLGLFL	318 ***	336 ***	281 ***	375 ***	243 ***	366 ***
FLGLFW	377 ***	153 ***	137 ***	428 ***	310 ***	76.9 ***
FLGLFA	327 ***	159 ***	226 ***	386 ***	225 ***	211 ***
Physiology						
EGC	91.1 ***	20.4 ***	44.8 ***	175 ***	59.6 ***	200 ***
NDVI_DTB	111 ***	31.1 ***	60.6 ***	79.4 ***	30.4 ***	97.2 ***
NDVI_DTHD	89.1 ***	32.9 ***	56.1 ***	82.5 ***	160 ***	60.9 ***
NDVI_Max	84.9 ***	17.1 ***	169 ***	74.1 ***	34.2 ***	0.047 ***
SR	470 ***	560 ***	130 ***	353 ***	398 ***	43.8 ***
CTIR	134 ***	162 ***	210 ***	84.4 ***	92.2 ***	21.5 ***
Yield and TGW						
TGW	687 ***	480 ***	670 ***	703 ***	581 ***	689 ***
GRYLD	116 ***	164 ***	224 ***	166 ***	561 ***	349 ***

*** *p* ≥ 0.001; †: PDG and PDL were considered traits of interest in season 2 and season 3.

**Table 2 genes-14-01507-t002:** The likelihood ratio test (LRT) was employed to assess the statistical significance of the model in the study. The use of the LRT is employed to determine the statistical significance of genotypic effects in the context of the genotype–environment interaction for MET analysis.

Traits	S1		S2		S3	
	LRTg	LRTge	LRTg	LRTge	LRTg	LRTge
Phenology						
DTB	101 ***	924 ***	146 ***	1120 ***	260 ***	318 ***
DTHD	302 ***	1220 ***	169 ***	1400 ***	202 ***	538 ***
DAYSMT	462 ***	97.2 ***	421 ***	66.4 ***	162 ***	176 ***
BTH	266 ***	127 ***	1.76 ns	318 ***	0.923 ns	285 ***
GFD	76.5 ***	797 ***	67.9 ***	1100 ***	221 ***	216 ***
PG_DTB	107 ***	890 ***	235 ***	852 ***	266 ***	334 ***
PG_DTHD	295 ***	882 ***	226 ***	1250 ***	249 ***	487 ***
PG_DAYSMT	496 ***	132 ***	466 ***	127 ***	173 ***	217 ***
PG_BTH	339 ***	114 ***	5.42 *	288 ***	27.3 ***	194 ***
PG_GFD	39.1 ***	283 ***	57.8 ***	276 ***	127 ***	44.7 ***
Plant Stature						
PH	379 ***	85.8 ***	239 ***	78.6 ***	31.4 ***	57.3 ***
SpkLng	231 ***	21.5 ***	146 ***	2.75 ***	193 ***	0.000 ns
PDG †	-	-	193 ***	54 ***	42.7 ***	51.3 ***
PDL †	-	-	238 ***	49.3 ***	109 ***	36.2 ***
HUS	380 ***	85.4 ***	235 ***	90 ***	30.5 ***	37.8 ***
FLGLFL	288 ***	31.9 ***	319 ***	25.7 ***	165 ***	114 ***
FLGLFW	374 ***	5.25 ns	249 ***	9.78 ***	43.8 ***	48.9 ***
FLGLFA	310 ***	17.7 ***	278 ***	6.04 ***	67.9 ***	132 ***
Physiology						
EGC	14 ***	98.2 ***	0.000 ns	84.3 ***	0.000 ns	249 ***
NDVI_DTB	20.3 ***	73 ***	3.69 ns	25.9 ***	5.43 *	72.4 ***
NDVI_DTHD	16.8 ***	59.4 ***	6.34 *	47.3 ***	1.34 ns	60.8 ***
NDVI_Max	33 ***	42.1 ***	10.9 ***	8.2 **	0.000 ns	132 ***
SR	217 ***	114 ***	293 ***	140 ***	0.000 ns	107 ***
CTIR	181 ***	0.818 ns	18.3 ***	83.7 ***	0.000 ns	94.6 ***
Yield and TGW						
TGW	646 ***	62.9 ***	202 ***	284 ***	412 ***	56.7 ***
GRYLD	106 ***	35.9 ***	1.42 ns	385 ***	143 ***	77.3 ***

*** *p* ≥0.001, ** *p* ≥0.01, * *p* ≥0.05; ns: non-significant; †: PDG and PDL were considered traits of interest in season 2 and season 3.

**Table 3 genes-14-01507-t003:** The statistical measures for the traits observed in early planting: mean ($\bar{x}$), confidence interval of mean (CI), and average deviation (AD).

Traits	Season 1			Season 2			Season 3		
	AD	CI	$\bar{x}$	AD	CI	$\bar{x}$	AD	CI	$\bar{x}$
Phenology									
DTB	4.59	0.318	88.1	7.07	0.47	92.80	3.96	0.29	89.70
DTHD	4.77	0.364	112	6.53	0.48	110.00	4.84	0.36	103.00
DAYSMT	2.53	0.181	158	2.41	0.18	165.00	3.10	0.23	162.00
BTH	2.99	0.215	24.1	3.02	0.22	17.60	1.93	0.14	13.70
GFD	3.44	0.267	45.8	4.99	0.37	54.50	3.17	0.23	58.40
PG_DTB	4730	327	53,000	6720	452	56,300	3450	259	50,000
PG_DTHD	7570	567	83,900	8010	577	75,300	4950	372	63,100
PG_DAYSMT	11,100	796	188,000	9120	663	169,000	9370	692	157,000
PG_BTH	970	70.3	3570	435	33	1370	214	17	734
PG_GFD	2270	171	20,900	2040	154	18,200	1820	136	20,400
Plant Stature									
PH	5.43	0.396	118	4.11	0.29	110.00	4.70	0.32	109.00
SpkLng	0.776	0.0557	11	0.85	0.06	10.50	1.18	0.08	12.90
PDG †	-	-	-	4.18	0.30	72.40	5.12	0.36	69.80
PDL †	-	-	-	3.17	0.23	37.60	2.75	0.19	39.40
HUS	5.24	0.38	107	3.93	0.28	99.40	4.61	0.32	96.30
FLGLFL	2.44	0.173	26.4	2.65	0.19	25.00	3.16	0.22	30.10
FLGLFW	0.116	0.0087	2.02	0.13	0.01	1.94	0.15	0.01	2.20
FLGLFA	5.21	0.374	40.1	4.95	0.354	36.5	7.33	0.527	50
Physiology									
EGC	3.3	0.238	84.3	3.21	0.27	80.10	6.07	0.44	67.10
NDVI_DTB	0.016	0.0012	0.802	0.02	0.00	0.78	0.03	0.00	0.81
NDVI_DTHD	0.0125	0.0009	0.796	0.02	0.00	0.76	0.02	0.00	0.81
NDVI_Max	0.0116	0.0008	0.835	0.02	0.00	0.81	0.02	0.00	0.85
SR	0.0011	0.0001	0.006	0.00	0.00	0.01	0.00	0.00	0.01
CTIR	0.0162	0.0011	0.179	0.01	0.00	0.13	0.03	0.00	0.21
Yield and TGW									
TGW	2.93	0.21	40.10	2.90	0.20	47.80	3.67	0.26	45.20
GRYLD	473.00	34.10	7580.00	597.00	42.90	7240.00	609.00	43.80	7920.00

†: PDG and PDL were considered traits of interest in season 2 and season 3.

**Table 4 genes-14-01507-t004:** The statistical measures for the traits observed in timely planting: mean ($\bar{x}$), confidence interval of mean (CI), and average deviation (AD).

Traits	Season 1			Season 2			Season 3		
	AD	CI	$\bar{x}$	AD	CI	$\bar{x}$	AD	CI	$\bar{x}$
Phenology									
DTB	1.80	0.13	89.70	2.37	0.18	103.00	3.61	0.27	101.00
DTHD	2.92	0.21	105.00	2.62	0.19	112.00	3.18	0.23	109.00
DAYSMT	2.89	0.20	145.00	1.80	0.13	153.00	2.28	0.16	156.00
BTH	2.48	0.18	15.10	0.97	0.07	8.71	1.65	0.12	8.57
GFD	1.82	0.13	40.10	1.68	0.12	41.20	1.96	0.14	46.30
PG_DTB	1830	131	45,500	3130	232	52,900	3840	282	48,800
PG_DTHD	5100	345	71,300	3680	263	64,700	4670	331	59700
PG_DAYSMT	11,100	770	156,000	7220	520	151,000	7770	551	145,000
PG_BTH	871	61	2890	128	9	602	190	15	578
PG_GFD	2440	170	15,800	1410	100	18,000	1510	106	18,600
Plant Stature									
PH	4.72	0.34	111.00	4.38	0.32	114.00	4.49	0.32	104.00
SpkLng	0.67	0.05	10.80	0.77	0.06	10.70	1.29	0.09	13.30
PDG †	-	-	-	4.15	0.29	76.00	4.50	0.32	66.90
PDL †	-	-	-	2.43	0.17	38.10	2.63	0.19	37.40
HUS	4.57	0.33	100.00	4.21	0.31	103.00	4.59	0.33	91.10
FLGLFL	2.36	0.17	25.90	2.20	0.16	23.80	2.95	0.21	30.20
FLGLFW	0.11	0.01	2.06	0.13	0.01	2.01	0.15	0.01	2.23
FLGLFA	5.09	0.36	40.2	4.72	0.342	36	6.81	0.484	50.7
Physiology									
EGC	4.06	0.30	82.80	6.47	0.45	75.30	8.87	0.63	70.30
NDVI_DTB	0.02	0.00	0.79	0.01	0.00	0.80	0.02	0.00	0.77
NDVI_DTHD	0.02	0.00	0.78	0.01	0.00	0.79	0.02	0.00	0.80
NDVI_Max	0.02	0.00	0.81	0.01	0.00	0.82	0.01	0.00	0.85
SR	0.00	0.00	0.01	0.00	0.00	0.01	0.00	0.00	0.01
CTIR	0.02	0.00	0.20	0.02	0.00	0.40	0.04	0.00	0.27
Yield and TGW									
TGW	2.82	0.20	40.40	3.45	0.25	40.40	3.43	0.25	42.20
GRYLD	501.0	35.9	7040.0	539.0	39.3	6860.0	601.0	42.9	7990.0

†: PDG and PDL were considered traits of interest in season 2 and season 3.

**Table 5 genes-14-01507-t005:** Selected traits and their increasing (up arrow in light-ash background) or decreasing (down arrow in light-brown background) selection pressures on multi-trait stability index analysis using MGIDI.

Traits	S 1	S 2	S 3	S 1	S 2	S 3
	EP	EP	EP	TP	TP	TP
PG_DTB	↑N	↑N	↑	↑	↑	↑
PG_BTH	↑	↑	↑	↓	↓N	↓N
PG_GFD	↑	↑	↑	↓N	↓N	↓N
HUS	↓	↓	↓N	↓	↓N	↓N
FLGLFL	↑	↑	↑	↑	↑	↑
FLGLFW	↑	↑	↑	↑	↑	↑
EGC	↑N	↑N	↑N	↑	↑	↑N
SR	↓	↓	↓	↓	↓	↓
CTIR	↓	↓	↓	↓N	↓	↓
TGW	↑N	↑N	↑N	↑	↑	↑N
GRYLD	↑	↑	↑	↑	↑	↑

N: Selection gain did not match desired selection pressure.

**Table 6 genes-14-01507-t006:** Distances of QTLs from *Vrn*, *Ppd*, and *Eps* loci along with nearest marker positions in the reference genomes.

Gene	Chromosome	Marker Position in the Reference Genome	Nearest Marker	Distance (Mbps)	QTLs	Controlling Traits		Reference
						*EP*	*TP*	
*Eps*	1D	485105000	63681750	421.42	QBth.bisa.1D.4		PG_BTH	[56]
			31127895	453.98	QMpt.bisa.1D.3	PG_GFD	PG_BTH	
*Eps*	1D	93484075	9364969	84.12	Qhus.bisa.1D.1		HUS	[57]
*Ppd-B1*	2B	56238081	59094836	2.86	QMpt.bisa.2B.2	SR, PG_BTH		[58]
*Ppd-D1*	2D	33955686	Inside the *QTL*	--	QMpt.bisa.2D.3	EGC, PG_GFD, PG_BTH	FLGLFW	[10]
*Vrn-A1*	5A	587423448	581738776	5.68	QMpt.bisa.5A.3	EGC, FLGLFL, HUS, PG_GFD, TGW	PG_BTH, FLGLFL	[8,59,60,61]
*Vrn-A3*	7A	71669854	71591808	0.08	QMpt.bisa.7A.4	GRYLD	PG_DTB	[62]
*Vrn-B1*	5B	573807893	576348143	2.54	QMpt.bisa.5B.3	SR, PG_DTB		[8,63]
*Vrn-B1*	5B	573807893	582830486	9.02	QMpt.bisa.5B.4	SR, PG_DTB, GRYLD		
*Vrn-B3*	7B	9702383	101512229	91.81	QMpt.bisa.7B.3	PG_DTB, SR, EGC	CTIR	[64]
*Vrn-B3*	7B	9702383	70866936	61.16	Qyld.bisa.7B.2	GRYLD		
*Vrn-D2*	4D	509341209	247154166	262.19	Qhus.bisa.4D.1	HUS		[65]
*Vrn-D3*	7D	68417074	68291897	0.13	QDtb.bisa.7D.3		PG_DTB	

The rows that are shaded indicate a high degree of proximity between the gene and the identified QTLs.

**Table 7 genes-14-01507-t007:** List of QTLs in different genomes significantly associated with different traits for early planting and timely planting.

**(a) Phenology**		
**Traits**	***QTL*s for EP**	***QTL*s for TP**
PG_DTB	QDtb.bisa.2D.4, QDtb.bisa.3B.1, QDtb.bisa.3B.3, QMpt.bisa.2A.3, QMpt.bisa.3B.2, QMpt.bisa.4B.2, QMpt.bisa.4B.6, QMpt.bisa.5A.1, QMpt.bisa.5B.2, QMpt.bisa.5B.3, QMpt.bisa.5B.4, QMpt.bisa.5B.5, QMpt.bisa.5D.1, QMpt.bisa.6A.3, QMpt.bisa.7B.3 = 15	QDtb.bisa.2D.4, QDtb.bisa.3B.5, QDtb.bisa.7D.3, QMpt.bisa.5B.2, QMpt.bisa.7D.2, QMpt.bisa.7A.4 = 6
PG_BTH	QBth.bisa.1A.1, QBth.bisa.2B.8, QBth.bisa.2B.9, QBth.bisa.3B.7, QBth.bisa.5A.2, QBth.bisa.7A.10, QBth.bisa.7A.11, QBth.bisa.7A.5, QBth.bisa.7D.3, QMpt.bisa.1B.3, QMpt.bisa.2B.2, QMpt.bisa.2D.3, QMpt.bisa.7A.6 = 13	QBth.bisa.1D.4, QBth.bisa.3B.6, QBth.bisa.4A.1, QBth.bisa.6A.2, QMpt.bisa.1D.3, QMpt.bisa.3A.5, QMpt.bisa.5A.3, QMpt.bisa.5B.1, QMpt.bisa.5B.2, QMpt.bisa.6B.2 = 10
PG_GFD	QBth.bisa.7A.5, QGfd.bisa.2D.2, QGfd.bisa.3A.2, QGfd.bisa.3D.2, QGfd.bisa.5D.2, QGfd.bisa.6A.4, QGfd.bisa.7A.5, QMpt.bisa.1D.2, QMpt.bisa.1D.3, QMpt.bisa.2A.3, QMpt.bisa.2A.6, QMpt.bisa.2D.3, QMpt.bisa.3A.5, QMpt.bisa.5A.1, QMpt.bisa.5A.3, QMpt.bisa.5B.2, QMpt.bisa.6A.1, QMpt.bisa.6A.3, QMpt.bisa.6B.2, QMpt.bisa.7B.5, QMpt.bisa.7D.2 = 21	QGfd.bisa.2B.3, QMpt.bisa.5A.1, QMpt.bisa.6A.3, QMpt.bisa.7D.2 = 4
**(b) Plant Stature**		
**Traits**	**QTLs for EP**	**QTLs for TP**
HUS	Qhus.bisa.4D.1, Qhus.bisa.5B.6, QMpt.bisa.2A.3, QMpt.bisa.5A.3, QMpt.bisa.6A.1, QMpt.bisa.6A.3, QMpt.bisa.6B.2, QMpt.bisa.7A.6 = 8	Qhus.bisa.1D.1, Qhus.bisa.2D.5, Qhus.bisa.3D.1, Qhus.bisa.7B.6, QMpt.bisa.6B.2, QMpt.bisa.7A.6, QMpt.bisa.7B.5 = 7
FLGLFL	QFlg.bisa.2A.2, QFll.bisa.3B.4, QFll.bisa.7A.7, QMpt.bisa.2A.1, QMpt.bisa.2A.6, QMpt.bisa.5A.3, QMpt.bisa.5B.1, QMpt.bisa.5D.1, QMpt.bisa.6A.3, QMpt.bisa.6A.5, QMpt.bisa.6B.2, QMpt.bisa.6D.2, QMpt.bisa.7A.6, QMpt.bisa.7D.2 = 14	QFll.bisa.2B.4, QFll.bisa.2B.6, QFll.bisa.3A.6, QFll.bisa.4A.2, QFll.bisa.6B.1, QFll.bisa.7A.1, QFll.bisa.7A.8, QFll.bisa.7A.9, QMpt.bisa.5A.3, QMpt.bisa.5B.1, QMpt.bisa.5B.2, QMpt.bisa.5D.1, QMpt.bisa.6A.3, QMpt.bisa.6B.2, QMpt.bisa.6D.3, QMpt.bisa.7A.6 = 15
FLGLFW	QFlw.bisa.2B.1, QMpt.bisa.2A.3, QMpt.bisa.6A.3 = 3	QFlw.bisa.1B.2, QFlw.bisa.2A.4, QFlw.bisa.5B.7, QMpt.bisa.1D.2, QMpt.bisa.2A.3, QMpt.bisa.2D.3, QMpt.bisa.6A.3 = 7
**(c) Physiological Traits**		
**Traits**	**QTLs for EP**	**QTLs for TP**
EGC	Qegc.bisa.7B.1, QMpt.bisa.2D.3, QMpt.bisa.5A.3, QMpt.bisa.6D.2, QMpt.bisa.6D.3, QMpt.bisa.7B.3, QMpt.bisa.7B.5 = 7	Qegc.bisa.3A.3, QMpt.bisa.3A.4 = 2
SR	QMpt.bisa.2A.1, QMpt.bisa.2A.3, QMpt.bisa.2B.2, QMpt.bisa.4B.6, QMpt.bisa.5B.1, QMpt.bisa.5B.3, QMpt.bisa.5B.4, QMpt.bisa.5B.5, QMpt.bisa.7A.6, QMpt.bisa.7B.3, QSr.bisa.7B.4 = 8	QMpt.bisa.5B.5, QMpt.bisa.5D.1, QMpt.bisa.6A.3, QMpt.bisa.7D.2, QSr.bisa.3A.1, QSr.bisa.7A.2 = 6
CTIR	none	Qcti.bisa.4B.5, QMpt.bisa.3B.2, QMpt.bisa.6A.3, QMpt.bisa.6B.2, QMpt.bisa.7A.6, QMpt.bisa.7B.3 = 6
**(d) TGW and Yield**		
**Traits**	**QTLs for EP**	**QTLs for TP**
TGW	QMpt.bisa.1B.3, QMpt.bisa.2A.3, QMpt.bisa.4B.2, QMpt.bisa.5A.3, QMpt.bisa.6B.2, QTgw.bisa.2A.5, QTgw.bisa.4B.1, QTgw.bisa.4B.3, QTgw.bisa.4B.4 = 6	QMpt.bisa.7A.6, QMpt.bisa.7D.2, QTgw.bisa.1B.1, QTgw.bisa.7A.3 = 4
GRYLD	QMpt.bisa.3A.4, QMpt.bisa.5B.4, QMpt.bisa.6A.5, Qyld.bisa.2D.1, Qyld.bisa.6D.1, QMpt.bisa.7A.4, Qyld.bisa.7B.2, Qyld.bisa.7D.1 = 8	QMpt.bisa.3A.4, QMpt.bisa.6B.2, Qyld.bisa.2B.5, Qyld.bisa.2B.7 = 4

## Data Availability

The datasets produced for this investigation can be obtained upon request from the first author and corresponding author.

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
