# Peer review of "GWAS for Early-Establishment QTLs and Their Linkage to Major Phenology-Affecting Genes (Vrn, Ppd, and Eps) in Bread Wheat"

_genes, 2023, doi:10.3390/genes14071507_

Round 1
Reviewer 1 Report
1- Introduction Section
please, discuss the previous studies that explain the role of candidate genes related to your traits of interest.
2- Figures legends are very abbreviated, please write more about the figures and add the statistics related to P val.
3- Manhattan plots is not clear, please move those figure as supplementary.
then add one or two clear Manhattan plots in the main manuscript.
4- English Grammar should be improved.
English Grammar should be improved.
Author Response
Dear Reviewer,
On behalf of the authors, I appreciate for the inputs and comments. The queries are addressed in the attached response letter and corresponding updates/revision in the manuscript.
Thanks and Regards

Reviewer 2 Report
Dear authors,
The manuscript entitled (GWAS for early establishment QTLs and their linkage to major 2 phenology-affecting genes (Vrn, Ppd and Eps)) included large set of data. However, this manuscript is so poorly crafted that is requires major revision
The title needs to be changed; it is not reflecting the importance of
The abstract lacking important clear information of the important findings of the manuscript. Please rewrite the abstract.
Arrange keywords in alphabetic order.
First, the manuscript is poorly written. Nearly every other sentence is confusing or illogical, contains words that do not quite "fit", has misspellings or other errors, etc.. Paragraphs are sometimes weakly constructed too (e.g., illogical flow, redundancy, unneeded information).
The English language must be revised by an expert or native language speaker.
For the a im of the work, arrange the aims beside each other's.
Line 100, what is the meaning of advanced spring wheat?
What is the meaning of wheat-growing window?
The material methods section needs to be improved where many parts I cannot understand.
What the meaning of stature? Many misunderstandable words must be revised.
What is the environmental conditions for plants growth?
Figure 3.1.1.e, the y axis is not readable.
Why the tables or figures named like Table 3.1.1.a, Figure 3.1.1.a? the authors can used table or figure1, 2, .....
The arrangement of tables is so difficult to be follow. some tables can be added in supplementary file and others gathered beside each other's.
The results section is simply unreadable. I strongly urge you to put all results in figure form, if you want a reader to be able to understand your results (with so many treatments/species/variables, putting so many of your results in tables is a nightmare for a reader). Figure legends and tables need units and improvements.
The authors must increase the resolution of Figure 3.1.6.a.
The discussion and conclusion must be improved.
The English quality is very poor and must be improved.
Author Response
Dear Reviewer,
On behalf of the authors, I appreciate for you inputs and comments. The queries are addressed in the attached response letter and corresponding updates/revision in the manuscript.
Thanks and Regards

Reviewer 3 Report
The research article titled " GWAS for early establishment QTLs and their linkage to major 2 phenology-affecting genes (Vrn, Ppd and Eps)" deals with identifying wheat lines for early establishment by analyzing agro-morphological characteristics and mapping genes or quantitative trait loci associated. The manuscript presented information about phenology affecting genes, including Vrn, Ppd and Eps. It is an important study with great significance because this region is important for wheat production in northern and central Indian regions. In summary, the study found that early seeding of wheat allowed for the selection of wheat lines that are tolerant to early high temperatures and have an extended phenological period. The "idiotype," which consists of increased photo-growing degree days for booting and heading, along with a longer grain filling period, is more suitable for early planting compared to timely planting. Additionally, early adapted idiotypes exhibited delayed senescence and a slower rate of canopy temperature rise, which are favourable traits. These findings suggest that a novel approach to wheat breeding should involve screening genotypes for early planting and designing idiotype with consistent and appropriate features. The manuscript clearly indicates the focus of the study, highlighting the use of GWAS for identifying early establishment QTLs and their connection to major phenology-affecting genes (Vrn, Ppd, and Eps). It effectively conveys the key elements of the research.
This manuscript has significance for the optimization of local germplasm resources. The manuscript required certain changes before further processing. Authors are recommended to make some minor corrections per suggestion.
1) Consider specifying the wheat, it would be beneficial to mention the specific plant varieties/ lines under investigation. This would provide readers with immediate context and relevance.
2) Authors can think on modification of titles as “An investigation the association between early establishment QTLs and key phenology-affecting genes (Vrn, Ppd, and Eps)
3) Clarify the term "early establishment QTLs": Since the term "early establishment QTLs" may not be familiar to all readers, it would be helpful to provide a brief explanation or definition in introduction introduction.
4) The abstract could benefit from rephrasing certain sentences to improve the flow and readability. For example, instead of "The current study focused on identifying wheat lines for early establishment by analyzing agro-morphological characteristics and mapping genes or quantitative trait loci associated with them," you could say "This study aimed to identify wheat lines suitable for early establishment through analysis of agro-morphological characteristics and genetic mapping of associated genes or quantitative trait loci
5) Provide context on BISA and Ludhiana, since it was appeared first time in text (Abstract)
6) Caption of table and figure should be standardised as per guidelines of journal.
7) Provide the details of code used in the study in supplementary file.
8) Authors should state the meaning of stars used in table below it.
9) AD and CI values are missing in table 3.1.1d, authors should explain possible reason and effect in results section.
10) Authors should redraw the figure in 3.1.1.e. the graph titles are overlapping.
11) Boxplot of phenology doesn’t show any units on Y axis.
12) Correct spelling of Venn diagram
satisfactory. need minor spelling corrections and some sentence corrections. Minor editing is required to get the true essence of the text.
Author Response

(The authors gave the same response as above.)

Reviewer 4 Report
The authors studied the development of new wheat idiotype adapted to earlier sowing date aiming to expand the time window for growing and increase grain yield. The classic and novel approaches are incorporated. The idea of the study is clearly explained and interesting. The manuscript is well written and organized. My comments and suggestions are inserted in the pdf version of the manuscript.

Author Response

(The authors gave the same response as above.)

Reviewer 5 Report
Abstract: Write the full name of BISA, because it is an abbreviation.
Abstract: line 25 – I think correct world is „ideotype“ not „idiotype“. The same line 26 + 18 times more.
Introduction: This is a good introduction but the adoption of early sowing may need to be made clearer.
Method: More information about the experimental formula can be added to make it easier for readers to understand.
The results: It is necessary to add more information about weather results in the results section because the goal of this article is to take advantage of the moisture of the soil for early sowing. Adding information to the results section because it is written generically.
Figures: Need to modify the figures, should have units, need to be consistent in size and colour of the font, and explain all abbreviations below the figure's body.
Tables: Need the format again because I think it is not following the guideline for the author and explain all abbreviations below the table body.
The manuscript is interresting and could be accepted after minor revision.
Author Response

(The authors gave the same response as above.)

Round 2
Reviewer 2 Report
The language should be checked.
I received a new version of the paper entitled "GWAS for early establishment QTLs and their linkage to major 2 phenology-affecting genes (Vrn, Ppd and Eps) in bread wheat", which I previously reviewed. Although I appreciate the efforts made by the authors, I still believe that this new version has major flaws preventing its acceptance. The authors have answered and they have handled all comments, but this is simply not true. In fact, the majority of the comments still occurs in this version.
I have highlighted the following concerns:
The pdf construction is not well performed. many sentences are missed and cannot be readable.
What is the change in the title?
The numbering of tables and figures is not acceptable.
The same presentation of figures.
The figures resolution still of low quality.
All captions are description of results. It should be rewritten.
Still the discussion and conclusion need to be improved.